# Direct observation of a superconducting vortex diode

Alon Gutfreund [1] ✉, Hisakazu Matsuki[2], Vadim Plastovets [3], Avia Noah [1], Laura Gorzawski[2], Nofar Fridman[1], Guang Yang[2], Alexander Buzdin [3], Oded Millo[1], Jason W. A. Robinson [2] ✉ & Yonathan Anahory [1] ✉

The interplay between magnetism and superconductivity can lead to unconventional proximity and Josephson effects. A related phenomenon that has recently attracted considerable attention is the superconducting diode effect, in which a nonreciprocal critical current emerges. Although superconducting diodes based on superconductor/ferromagnet (S/F) bilayers were demonstrated more than a decade ago, the precise underlying mechanism remains unclear. While not formally linked to this effect, the Fulde–Ferrell–Larkin–Ovchinikov (FFLO) state is a plausible mechanism due to the twofold rotational symmetry breaking caused by the finite center-of-mass-momentum of the Cooper pairs. Here, we directly observe asymmetric vortex dynamics that uncover the mechanism behind the superconducting vortex diode effect in Nb/EuS (S/F) bilayers. Based on our nanoscale SQUID-on-tip (SOT) microscope and supported by in-situ transport measurements, we propose a theoretical model that captures our key results. The key conclusion of our model is that screening currents induced by the stray fields from the F layer are responsible for the measured nonreciprocal critical current. Thus, we determine the origin of the vortex diode effect, which builds a foundation for new device concepts.

In recent years, a significant theoretical and experimental effort has been directed toward the interplay between magnetism and superconductivity[1–13]. Incorporating spin-orbit coupling (SOC) and considering the presence of a magnetic exchange field can theoretically predict the emergence of a nonreciprocal critical current[14]. These predictions have been realized experimentally as the superconducting diode effect in superconductor/ferromagnet (S/F) bilayers[15–17]. In such a device, the critical current is not symmetric with respect to the applied current direction and depends on magnetization. Understanding the underlying physical mechanism and its relation to vortex motion is essential to harness the diode effect for applications in superconducting electronics.

Conventional (*s*-wave) superconductivity is mediated by spin-singlet Cooper pairs in which each electron of a pair has the opposite sign of spin. Ferromagnetism favors a parallel alignment of electron spins and so the proximity effect at an S/F interface suppresses superconductivity[18–23] and breaks time-reversal symmetry, establishing the Fulde–Ferrell–Larkin–Ovchinikov (FFLO) state[24,25].

In a system that exhibits Rashba SOC[26], the spin bands split, and a ferromagnetic exchange field acts differentially on the oppositely aligned electron spins within the Cooper pairs[27]. Hence the Cooper pairs gain a nonzero center-of-mass-momentum that is magnetization-orientation-dependent[14,28,29]. This type of symmetry breaking has been observed in S/F bilayers as a nonreciprocal critical current[17,30–36] and an

[1]The Racah Institute of Physics, The Hebrew University of Jerusalem, Jerusalem 9190401, Israel. [2]Department of Materials Science & Metallurgy, University of Cambridge, Cambridge CB3 0FS, United Kingdom. [3]LOMA UMR-CNRS 5798, University of Bordeaux, Talence F-33405, France. ✉e-mail: alon.gutfreund@mail.huji.ac.il; jjr33@cam.ac.uk; yonathan.anahory@mail.huji.ac.il

asymmetric intermediate state of resistance[15,16]. Despite recent theoretical progress[14,37], the connection of these observations to the microscopic explanation remains unclear.

In the present study, we focus on the origin of the diode effect in bilayers of EuS/Nb. Using magnetic imaging techniques, we demonstrate asymmetric vortex dynamics that are manipulated by the magnetization of the ferromagnet and correspond with a nonreciprocal critical current. While the role of vortex dynamics is largely overlooked in the literature regarding the diode effect, the unidirectional trajectories of vortex motion that we observe suggest an alternative underlying mechanism. These surprising results can be explained by taking into account the screening current distribution induced by an inhomogeneous magnetic field emanating from the ferromagnet.

## Results
### Magnetic response to an ac current
The device consists of a EuS/Nb bilayer patterned into a Hall bar (Fig. 1a; see Methods). We assume that there should be no effect on the magnetic properties of the EuS film due to the coupling with superconducting Nb. We verify this assumption by performing global magnetization measurements on an unpatterned EuS/Nb device with identical thicknesses, which show a negligible difference across the SC transition temperature (Supplementary Fig. S1). We apply an ac current $I_x^{ac}$ along the EuS/Nb wires and simultaneously record the magnetic field response that is modulated in phase with the current using the SOT. In order to generate free vortices in the sample, an out-of-plane (OOP) field $|\mu_0 H_z| = 5$ mT was applied in all the measurements presented in this work. In Fig. 1b, we show a $B_z^{ac}(x,y)$ image that corresponds to a longitudinal voltage $V_x = 0$, meaning that the current amplitude is smaller than the critical current $I_c$. The image shows the expected features for a Biot–Savart field induced by a current flowing in a superconducting slab[38,39]. For $I_x^{ac} > I_c$ ($V_x \neq 0$), lobe-shaped features appear in the image at the sample edges (see Fig. 1c). In this regime, the

intensity of the observed features grows linearly with the applied transport current amplitude until the switching current ($I_s$) is crossed and the sample transitions to a normal state. We note that a positive signal (blue on the colormap) means that the magnetic feature in the time domain appears in phase with respect to the applied current, whereas a negative signal (red on the colormap) means that there is a π-phase difference with respect to the current. Since these features appear with the onset of voltage, it is reasonable to assume that they are the result of vortex flow. Such vortex channels were already observed by applying a dc current[39]. The channels that are in phase (π-phase) with respect to the current appear when the instantaneous current is $I_x^{ac}(t) > 0$ ($I_x^{ac}(t) < 0$).

Vortex channels are observed only along the sample edges due to bifurcations that occur along their path, which randomize their trajectories. The bifurcation originates from the nonuniform current distribution that modulates along the $y$-axis[38]. The vortex enters the sample with a higher velocity due to the larger current density along the edge and slows down as it penetrates into the sample where the current density is lower. Finally, as the vortex flows toward the center, the vortex–vortex distance decreases, and the mutual repulsive force causes the vortices to bifurcate[39]. Given that we average over an ac signal with a time scale many orders of magnitude larger than those involved in the vortex dynamics, our images effectively portray an averaging of vortex paths across the device. For that reason, the signal along the channels becomes undetectable in the inner part of the sample. We emphasize that we are observing vortices in a thin film superconductor with an effective penetration depth determined by the Pearl length, $\Lambda \sim 6\,\mu$m. This length scale defines the effective size of the vortex. Having a larger vortex implies that the same magnetic flux is spread out over a larger area, resulting in weaker fields that are harder to detect. This further smears the vortex signal, hampering the observation of single vortex channels.

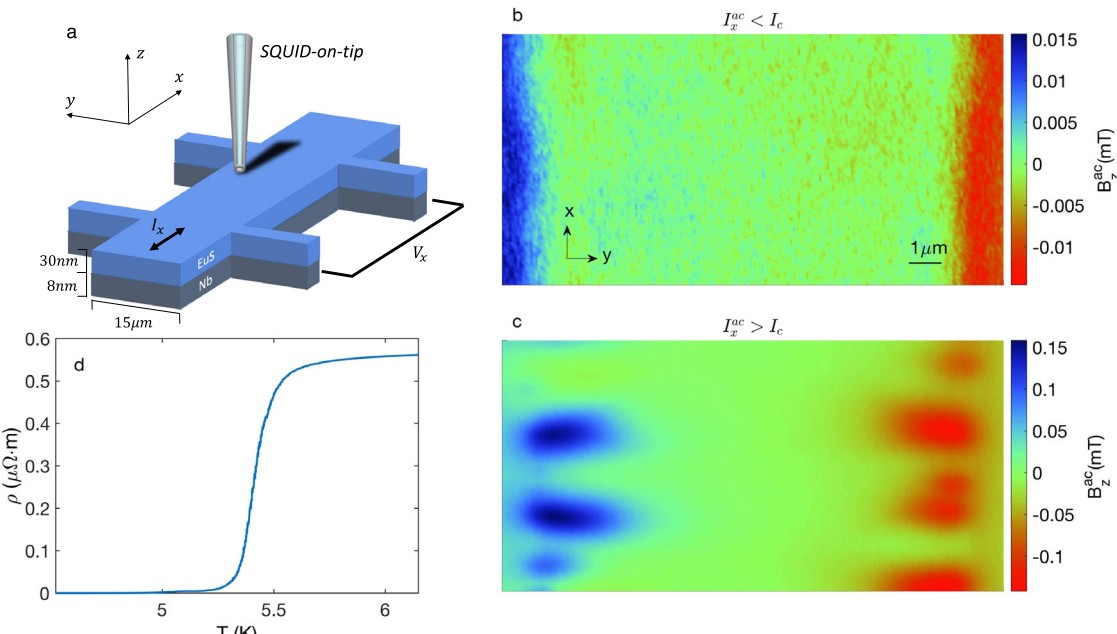

**Fig. 1 | Magnetic response of the EuS/Nb bilayer to an ac current $I_x^{ac}$. a** Schematic diagram of the measurement setup, showing a EuS/Nb Hall bar structure, along with the scanning SQUID-on-tip (SOT) probe. **b** SOT image of the ac out-of-plane (OOP) component of the magnetic field $B_z^{ac}(x,y)$ modulated with respect to an oscillating transport current with root-mean-square (RMS) value $I_x^{ac} \simeq 0.15$ mA $< I_c$. Blue (red) corresponds to a positive (negative) OOP component of the field emanating from the Nb strip. The device was zero-field cooled to 4.2 K,

below the superconducting transition ($T_c \sim 5.5$ K) and Curie temperature ($T_C \sim 20$ K). **c** Same as (**b**) but with RMS value $I_x^{ac} \simeq 0.42$ mA $> I_c$. In this case, the polarity of the signal depends on whether the magnetic feature appears in phase (blue) or at a π-phase (red) with respect to the oscillating current. The value of the measured magnetic field can be obtained from the color bar on the right side of the images. **d** $R(T)$ measurements of the device showing the superconducting transition.

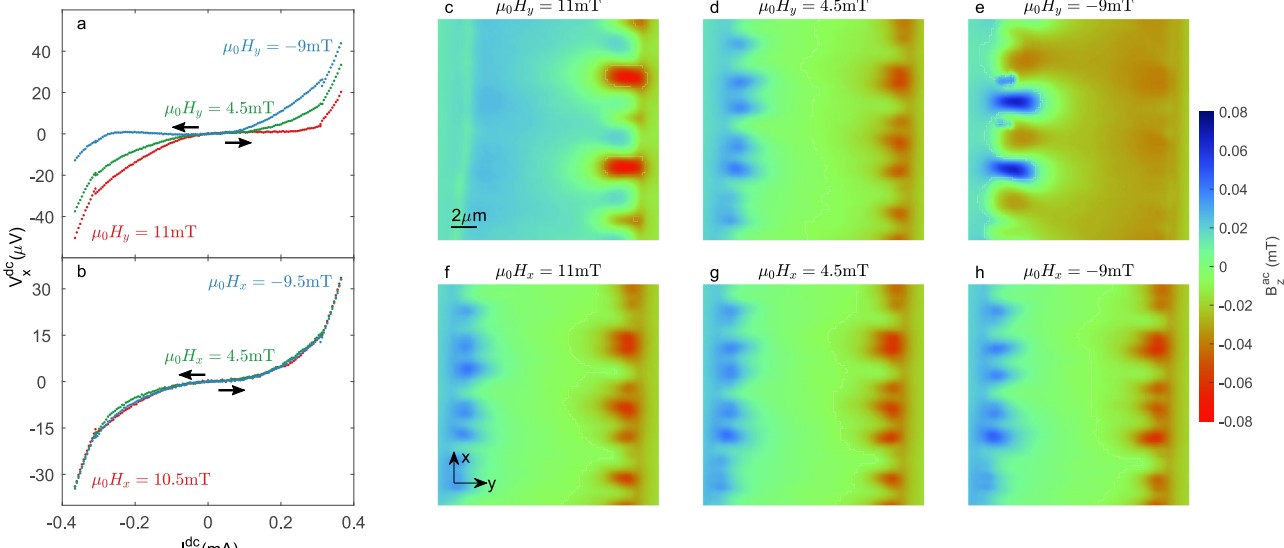

**Fig. 2 | Vortex flow and corresponding transport measurements as a function of an applied in-plane magnetic field. a, b** $I(V)$ characteristics for different transverse (**a**) and longitudinal (**b**) magnetic fields; arrows indicate the direction of the current sweep. The red and blue curves are at fields beyond the saturation field ($H_s$), while the green curve is at the coercive field. Black arrows indicate that the transport curves were always swept from zero to maximum bias in order to eliminate the effect of hysteresis caused by heating. **c–h** SQUID-on-tip (SOT) image of the ac out-of-plane component of the magnetic field $B_z^{ac}(x,y)$ modulated with respect to an oscillating transport current with an RMS value $I_x^{ac} \simeq 0.42\text{mA} > I_c$. The polarity of the signal depends on whether the magnetic feature appears in phase (blue) or at a π-phase (red) with respect to the oscillating current. **c–e** Transverse magnetic field orientation ($H_y$) with $\mu_0 H_y$ above $H_s$ in the +$y$ direction (**c**), −$y$ direction (**e**), and at the coercive field (**d**). **f–h** Same values of the magnetic field but in a longitudinal orientation, parallel to the direction of current $H_x \parallel I_x^{ac}$. The value of the measured magnetic field can be obtained from the color bar on the right side of the figure. The indicated color scale is the same for all images. For a full set of $B_z^{ac}(x,y)$ images, see Supplementary movies 1–2.

## Magnetization-dependent vortex flow

We first discuss the case where we fully magnetize the EuS by applying an in-plane (IP) magnetic field in the $y$ direction ($\overrightarrow{\mathbf{H}} \parallel \hat{\mathbf{y}}$), perpendicular to the current flow, $H_y = 11$ mT. The magnetic image $B_z^{ac}(x,y)$ acquired in that state is shown in Fig. 2c. Unlike what is observed in the ZFC state (1c), the vortex channels are now visible only on the right edge of the sample. Interestingly, once the EuS is fully magnetized in the −$y$ direction, by applying $H_y = -9$ mT, vortices penetrate only from the opposite, left edge (Fig. 2e). This suggests that as a result of the sample magnetization, vortices only penetrate into the sample from one edge. Around the coercive field ($H_c = 4.5$ mT), the anti-symmetric image observed in the ZFC state is recovered (Fig. 2d). These results clearly demonstrate the emergence of a vortex diode effect in which the diode direction is set by the magnetization and vanishes around $H_c$.

We now turn to discuss the case where the magnetization is in the $x$ direction ($\overrightarrow{\mathbf{H}} \parallel \hat{\mathbf{x}}$), parallel to the current. In this case, the vortex channels appear on both edges regardless of whether the magnetization is in the +$x$ direction (Fig. 2f), −$x$ direction (Fig. 2h), or at the coercive field (Fig. 2g). These results indicate that the vortex diode effect vanishes when the magnetization is parallel to the current or that there is no net magnetization.

The observation that vortices enter with more ease from one edge compared to the other implies that the critical current in the positive direction should be different from that in the negative direction ($|I_c^+| \neq |I_c^-|$). To confirm the absence of a reciprocal critical current, we pass a dc current $I_x^{dc}$ while measuring the longitudinal voltage $V_x$ for different magnetization directions. The results for the case where the IP magnetization is perpendicular to the current are depicted in Fig. 2a. The red curve was measured while applying $H_y = 11$ mT and corresponds to the SOT image shown in Fig. 2c. A clear asymmetry is observed between the sweep from zero to maximum positive current compared to the opposite direction, in accordance to the unidirectional vortex flow image shown in Fig. 2c. The asymmetry is reverted by changing the magnetization direction (blue curve, acquired under

$H_y = -9$ mT), consistent with the reversed vortex flow direction (Fig. 2e). Importantly, no asymmetry in the $I(V)$ characteristics is observed around the coercive field (green curve), consistent with vortices penetrating from both edges (Fig. 2d). In the case where the field is applied along the $x$-axis (2b), the curves are nearly independent of the applied field and give nearly symmetrical values of $I_c$, consistent with the images (Fig. 2 f–h).

## The origin of the diode effect

To better quantify the asymmetry, we plot the difference in the absolute value of the critical current in each direction ($\Delta I_c = |I_c^+| - |I_c^-|$) versus the applied magnetic field (Fig. 3a; see Methods). Further clarification on the definition of $I_c$ in this context is presented in Supplementary note 1. For $H_y$, three states are visible. $\Delta I_c$ is positive when the magnetization is in the +$y$ direction, $\Delta I_c$ is negative when the magnetization is in the −$y$ direction, and $\Delta I_c \sim 0$ around $H_c$. The values of $H_c$ (about 5 mT) revealed by the diode effect are consistent with the volumetric magnetization measurement of the unpatterned bilayer of matching thicknesses (Fig. 3b). This confirms that the direction of the vortex diode depends on the magnetization orientation and that when the net magnetization vanishes so does the diode effect. From the images acquired for different longitudinal fields $H_x$, we expect no diode effect at any applied field; indeed, we find $\Delta I_c \sim 0$ for all values of $H_x$.

We now turn to the origin of the vortex diode effect by looking at the magnetic texture of the EuS. The OOP component of the static magnetic field $B_z^{dc}(x,y)$ was acquired as a function of the applied IP magnetic field (Fig. 3c–e). In these SOT images, only the stray field generated by the magnetization of the EuS is visible. For a fully magnetized sample along the $y$ direction, large stray fields are observed at the edges. For $H_y > +H_s$, the field enters on the left edge and exits on the right (see Fig. 3c). The direction of the field lines is reverted once the magnetization is reversed (Fig. 3e). At $H_y \sim H_c$, the range in $B_z^{dc}(x,y)$ is significantly smaller, by roughly a factor 12 (note the different color

scale between the images), and shows a disordered structure. The observed magnetic correlation length $\xi_f$ is on the order of the SOT diameter, ~200 nm, suggesting that the domain size is even smaller. The same type of images showing disordered, low-magnitude magnetic structure are observed for all applied $H_x$ (see Fig. 3f–h). In this case, the large stray field in the $z$ direction appears only at the $x$ boundaries of the sample (outside the Hall bar). Therefore, in the region of interest, the field range is small in magnitude, and only a disordered magnetic structure is observed.

## Theoretical model

The quantitative observations we show in $B_z^{dc}(x,y)$ are transferred to a theoretical model in order to explain the origin of the vortex diode effect. Consider an infinite S/F bilayer strip as schematically sketched in Fig. 4a. We assume that the F layer is fully magnetized along the $y$ direction, and the corresponding homogeneous magnetization $4\pi M$ induces a stray field $\overrightarrow{H}$ in the $zy$ plane. The corresponding vector potential $\overrightarrow{A}_M$ can be derived from the magnetostatic equivalent of the Poisson equation, assuming the magnetic sources are two infinite auxiliary wires located on the edges of the F layer (Fig. 4a)[40]. The stray field from F is screened by the supercurrent in S. The density of the screening current, which subsequently affects vortex flow, can be found within the framework of the London approach.

$$\overrightarrow{j}_s(y) = -\frac{c}{4\pi\lambda^2}\left[\overrightarrow{A}_s(y) + \overrightarrow{A}_M(y) + \overrightarrow{A}_0\right], \quad (1)$$

where $\overrightarrow{A}_s$ is the vector potential induced in S and can be found from Biot–Savart's law. $\overrightarrow{A}_0$ is a gauge term that imposes $\int(\overrightarrow{j}_s(y)\cdot\hat{x})dy = 0$. The resulting implicit equation for $\overrightarrow{A}_s(y)$ determines the distribution

of the screening current density $j_s(y)$ in the superconducting film and can be solved iteratively[41].

An analytic expression for the transport current distribution can be obtained from a modification of the Bean critical state model[38,42]:

$$j_x(y) = \begin{cases} \frac{2j_c}{\pi}\arctan\left(\sqrt{\frac{(L/2)^2-a^2}{a^2-y^2}}\right), & \text{if } |y| < a \\ j_c, & \text{if } a < |y| < L/2, \end{cases} \quad (2)$$

where we define the parameter $a = \frac{L}{2}\sqrt{1-\left(\frac{I_t}{I_c}\right)^2}$, which can be interpreted as half of the central field-free region. $I_t$ is the total transport current, and $I_c$ and $J_c$ are the critical current and the critical current density, respectively.

In Fig. 4b, we show a comparison between the numerical calculation of the screening current density and the analytical expression for the transport current density. A more detailed derivation of the model, along with a discussion about the validity of the theoretical assumptions, is provided in supplementary note 2.

It is important to note that the absolute value of the screening current density $j_s$ is larger than that of the transport current density $j_t$ along the edges, but it is smaller in the bulk. Therefore, in this case, the total current density $j_{tot}^-$ (Fig. 4c, red curve) would be of the same sign throughout the whole length of the device, permitting flux flow. On the other hand, if we switch the direction of the transport current, it will tend to cancel with the screening current at the edges, resulting in a significantly lower total current in absolute value $j_{tot}^+$ (blue curve). When the total current is far below the depairing limit on the edge, new vortices cannot penetrate, thus preventing flux flow. Moreover, the total current density must cross zero at two points near the edges,

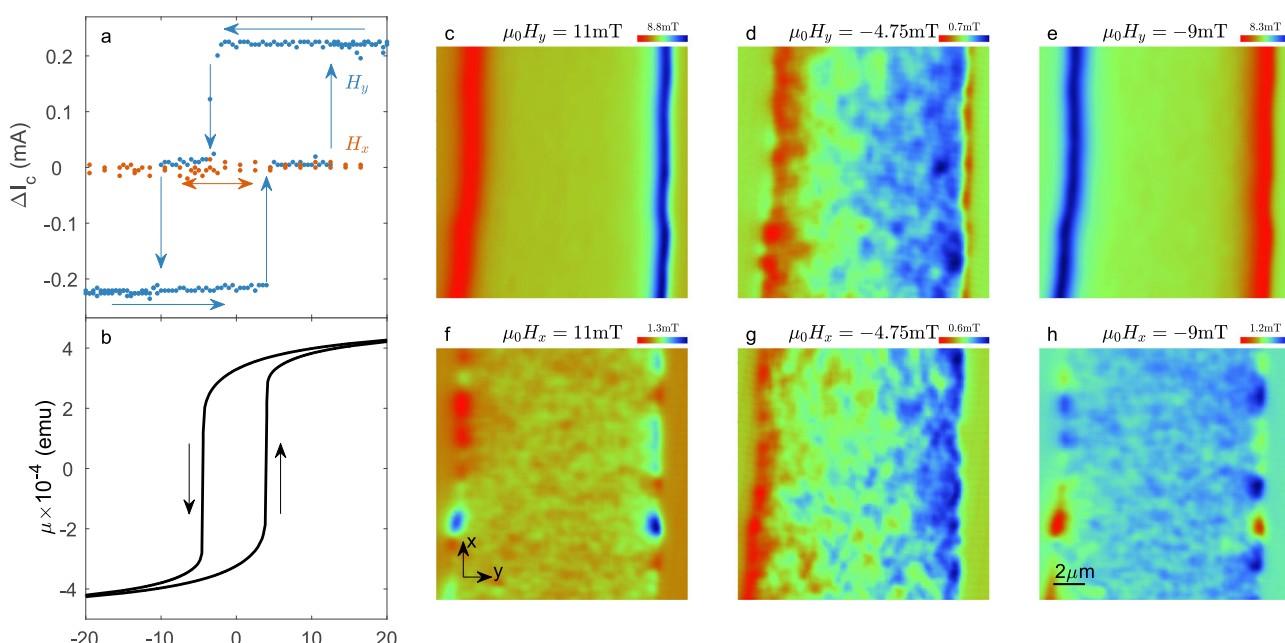

**Fig. 3 | Correlation between the vortex diode effect and the magnetic texture of EuS. a** Asymmetry factor $\Delta I_c = |I_c^+| - |I_c^-|$ as a function of magnetic field for transverse (blue symbols) and longitudinal (red symbols) magnetization. The arrows mark the magnetic field sweep directions. **b** n-plane M(H) curve of an unpatterned EuS/Nb film with the same thicknesses as our device. **c–h** SQUID-on-tip (SOT) images of the static out-of-plane (OOP) component of the magnetic field $B_z^{dc}(x,y)$ emanating from the EuS/Nb bilayer under various applied in-plane (IP) magnetic

fields, as indicated. **c–e** Transverse ($H_y$) magnetic field orientation with $\mu_0 H_y$ above the saturation field in the +$y$ direction (**c**), −$y$ direction (**e**) and at the coercive field (**d**). **f–h** Same values of magnetic field but in an longitudinal orientation ($H_x$). The signal range (in mT) for each individual image is noted above the color bar that appears in the top right corner of the image. The center of the scale (green color) is calibrated to the external OOP magnetic field $\mu_0 H_z = -5$ mT. For a full set of $B_z^{dc}(x,y)$ images, see Supplementary movies 1–2.

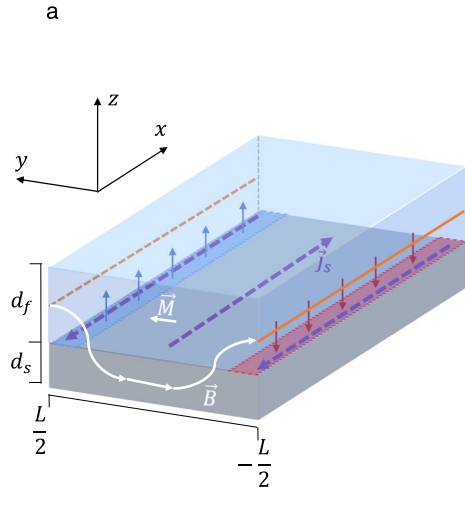

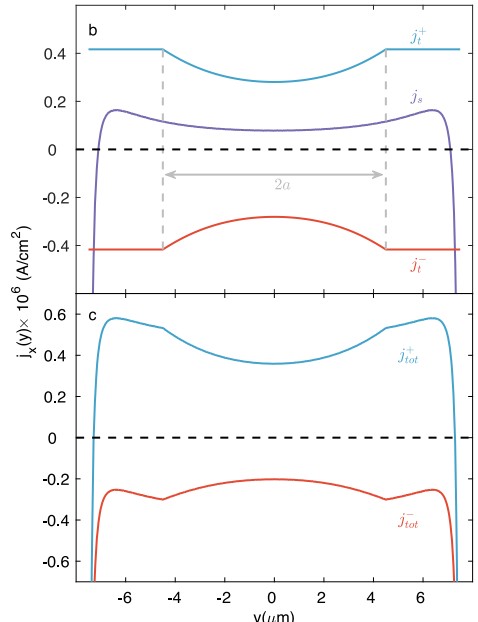

**Fig. 4 | Theoretical model. a** Schematic diagram of an S (gray)/F (blue) bilayer fully magnetized along the $y$ direction. The orange lines represent auxiliary wires (at $y = \pm L/2$, $z = 0$) with linear magnetic charge density $\pm Md_f$, which generate a stray magnetic field $\vec{\mathbf{H}}$. This field induces a screening supercurrent $j_s(y)$ inside the S film (purple dashed lines) that generates a field in the opposite direction (red and blue arrows). The sum of the fields ($\vec{\mathbf{B}}$) generated by the S and F layers in the $zy$ plane is represented by the white line. **b** Calculation of the transport current densities at opposite phases of the ac cycle (blue and red curves) along with the screening current density (purple curve). The central field-free region $2a$ is depicted by the region between the dashed gray lines. **c** Total current density $j_{tot}$ (transport + screening) at opposite phases of the ac cycle. In these calculations, the value of $I_t/I_c$ (which determines the value of $a$) was set to 0.8, implying that the system is approaching the critical current.

confining the vortices to these regions while prohibiting flux flow. Apart from the claim that the entry barrier of the sample is overcome, another condition that needs to be fulfilled is that the barrier to leave the sample must be overcome as well. Our data suggest that the Meissner screening currents caused by the EuS have the effect of lowering both barriers. For a given saturated magnetization, vortices are repelled from the entry point and attracted by the exit point. That is what favors one direction of vortex flow. Furthermore, the existence of the diode effect, as observed both in transport and in SOT microscopy, contradicts the dominance of pinning in the bulk of the sample, as pinning should suppress the possibility of flux flow, regardless of the entry and exit barriers.

## Discussion

The key aspect of the model is that in the presence of stray fields from the F layer, the distribution of screening currents in the S layer enhances flux flow in one direction of transport current and limits it in the other direction. The direction of the screening currents can be reversed by inverting the magnetization direction and can also be significantly suppressed by initiating a magnetically disordered state in the F layer (i.e., by applying magnetic fields around the coercive field or by applying a field along the $x$ direction). This simple explanation captures the key aspects of our results and does not require the presence of an FFLO state. Therefore, the diode effect could be observed regardless of the SOC strength. This is consistent with recently reported data that shows comparable diode efficiencies in samples with highly different SOC strength[17]. However, we stress that in the case where vortex dynamics is not involved, an FFLO state and large SOC strength seem to be required for the diode effect to emerge[31]. In our device, we observe ultrafast vortex dynamics with frequencies on the order of 1 GHz. A rough estimate of this value is given by considering the voltage produced by one vortex channel $V = \Phi_0 f$, where $\Phi_0$ is the flux quantum, and $f$ is the frequency of the channel. From our images (Fig. 2), we estimate roughly 0.3 vortex channels per µm. Given

the geometry of our sample, this implies that we have ~20 vortex channels between the voltage electrodes. Finally, taking the typical voltage obtained by our transport measurements (40 µV) (Fig. 2a), we can calculate $f \approx 9.6 \times 10^8$ Hz. The predicted nonreciprocal critical current that triggers these highly dissipative vortex flows is observed in both local imaging techniques and global transport measurements. We therefore provide deeper insight into the underlying mechanism responsible for the superconducting diode effect. This progress should enable the development of reliable, tunable devices that can act, for example, as high-frequency voltage rectifiers in superconducting electronics.

## Methods
### Sample fabrication
EuS/Nb bilayers were fabricated by ultra-high vacuum electron beam evaporation at room temperature with a base pressure of better than ~$1 \times 10^{-8}$ Torr. The main sample measured, whose results we present here, consists of a 30 nm thick layer of EuS on an 8 nm thick layer of Nb evaporated on a substrate of SiO$_2$. The EuS is capped with a 3 nm thick layer of non-superconducting Nb. The EuS has a strong IP anisotropy with a saturation field of $H_s \approx 10$ mT that is independent of the IP field angle[43]. The superconducting critical temperature of this sample is ~5.4 K. We present global magnetization measurements of an unpatterned EuS/Nb device with the aforementioned thicknesses in Supplementary Fig. S1.

### Transport measurements
Transport measurements were carried out at 4.2 K inside a liquid helium dewar employing a standard four-probe configuration, where the distances between the voltage contacts were 67.5 µm. Current sweeps were performed from 0 to ±0.5 mA. A dual-axis magnet consisting of standard SC coils was used to apply both OOP and IP magnetic fields. The probe itself was rotated in the sample space to control the angle of the applied IP field.

## SOT fabrication

The quartz tubes were pulled using a Sutter Instrument P2000 micropipette puller to create tips with a diameter of ~160 or ~250 nm. The SOT was then fabricated using self-aligned three-step thermal deposition of Pb at cryogenic temperatures. In the first step, the pipette was pointed toward the source, and a thin film was deposited onto the apex ring of the pipette, forming the superconducting loop of the SQUID. The pipette was then rotated to a 100° orientation, and an electrode was deposited on one side of the pipette, connecting the apex ring and the gold contact. The third deposition was performed at a −100° orientation, forming the second electrode on the opposite side of the pipette[44,45]. The carefully adjusted deposition thicknesses resulted in SQUIDs with a critical current ranging from 60 to 120 µA at zero field. The relatively large diameter tip allows for high magnetic field sensitivity, and a slight asymmetry in the Josephson junctions shifts the interference pattern of the SQUID, resulting in finite magnetic field sensitivity at low applied fields[44]. This is crucial in order to conduct the experiment at a low enough OOP field to avoid overcrowding the sample with vortices.

## Scanning SOT measurements

The sample was ZFC from room temperature to 4.2 K, below the critical temperature ($T_c$ = 5.4 K, see Fig. 1d). An alternating current $I_x^{ac}$ at $f \approx 1.1$ kHz was imposed along the $x$-axis. Simultaneously, using the SOT, the OOP component of the magnetic field that is modulated in phase with the current, $B_z^{ac}(x,y)$ is recorded. In order to generate free vortices in the sample, an OOP field $|\mu_0 H_z| = 5$ mT was applied in all the measurements presented in this work.

## Data analysis

In Fig. 3a, where we present the asymmetry factor $\Delta I_c$, $I_c$ is defined as the lowest current in which we measure an onset of voltage. To determine this point, we take the derivative of $V(I)$ and impose a threshold above the noise level of 20 mΩ. Supplementary Fig. S2 shows the derivative of a typical $V(I)$ curve with the corresponding threshold of 20 mΩ that was used to obtain the data in Fig. 3a. It is also possible to perform this analysis by assuming a threshold on the voltage value of the curves. For this method, we impose a threshold of 2 µV. We obtain similar results using this criterion, as shown in Supplementary Fig. S3.

## Data availability

The data that supports the findings of this study have been deposited in the GitHub database: https://github.com/QIL123/EuSNb.

## Code availability

The MATLAB scripts that analyze the raw data and reproduce the figures appearing in this paper have been deposited in the GitHub database: https://github.com/QIL123/EuSNb.

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

## Acknowledgements

We thank O. Agam, D. Orgad, E. Zeldov, and A. Kamra for fruitful discussions. This work was supported by the European Research Council (ERC) Foundation Grant No. 802952 (Y.A.), the EPSRC through the Core-to-Core International Network "Oxide Superspin" (Grant No. EP/P026311/1) (J.W.A.R.), the French National Agency for Research, Idex Bordeaux (Research Program GPR Light) and the EUR Light S&T (V.P. and A.B.). The international collaboration on this work was fostered by the EU-COST Action Nanocohybri CA16218 and Superqumap CA21144.

## Author contributions

A.G., Y.A., O.M., and J.W.A.R. conceived the experiment. A.G. realized the SOT experiment. A.G., O.M., J.W.A.R., and Y.A. analyzed and interpreted the data. A.G., A.N., and N.F. fabricated the SOT sensor. H.M., L.G., and G.Y. fabricated the sample and performed the volumetric magnetization measurements. V.P. and A.B. conceived the theoretical model. A.G., O.M., J.W.A.R., and Y.A. wrote the manuscript with input from all the co-authors.

## Competing interests

The authors declare no competing interests.
