## [Peer Review File · Nature Communications]

Direct Observation of a Superconducting Vortex DiodeREVIEWER COMMENTS

Reviewer #1 (Remarks to the Author):

The idea of direct imaging a superconducting vortex diode effect is very interesting. However, the present results are not convincing. This is because there are no images showing clear vortices and/or vortex flow channels. In contrast, unambiguous vortices and vortex flow channels were demonstrated in Ref. 30 which was published by one of the same corresponding authors and using the same imaging technique. I guess it could be difficult to directly visualize the vortex flow channels on the present sample with a magnetic layer on top of a superconducting film. The EuS could be a magnetic shielding layer that would prevent the accessing of the magnetic signals from vortices. I suggest the authors to repeat the experiments on a sample in which the magnetic EuS layer is under the superconducting film. In this case, the vortices and the diode-like vortex flow could be directly visualized.

Reviewer #2 (Remarks to the Author):

This is a very thorough and impressive experimental work showing how a ferromagnetic layer modifies the transport properties of a superconductor through its influence on vortices (fluxes). The results are of broad interest given the current trends in superconducting electronics for energy efficient and quantum information technologies. One aspect, I believe, is not described clearly in the paper: what is the behaviour of the magnetic layer without the superconductor? Which images were taken above T_c (if any)? If there are no such data, can the authors describe the expected behavior of the FM layer only?

Reviewer #3 (Remarks to the Author):

The present manuscript describes combined scanning SQUID and transport experiments in superconductor/ferromagnet bilayers. The goal is to elucidate the mechanism leading to the emergence of nonreciprocal supercurrent (superconducting diode effect) in ordinary superconductors when placed on ferromagnets.

On the one hand, this mechanism is here demonstrated to be pretty conventional, as anticipated by other works (Ref.[17], but also by [D. Suri et al., Applied Physics Letters 121, 102601 (2022).]). In addition, the experimental technique is not totally new. Still, the application of this imaging technique to this particular problem helps to clarify directly and beyond any doubt the source of nonreciprocal supercurrent in S/F bilayers. Owing to the great interest in this topic by the scientific community, I find the manuscript potentially suitable for publication in Nature Communications, provided that the questions below are addressed.

1) The authors write in the introduction:

"This type of symmetry breaking has been observed in S/F bilayers as a non-reciprocal critical current [15–19] and..."

The authors here are talking about a system with Rashba SOC with additional exchange interaction (see beginning of the paragraph). The references 18-19 appear therefore misplaced in this context, since there is no SF bilayer there. If, instead, the goal of the authors is simply to refer to the recent observations of spin-perturbation induced SC diode effect (which I would find appropriate for this paper), then 18-19 are fine but then I would, for consistency, also add two

relevant works that appeared before Ref 18, namely:
-C. Baumgartner, et al., Nature Nano. 17, 39 (2022).
-H. Wu et al., Nature, 604, 653 (2022).

Also, given the topic treated in this manuscript, I would recommend to cite the aforementioned reference [D. Suri et al., Applied Physics Letters 121, 102601 (2022).]

2) Sometimes symbols and acronyms are defined first in Methods and not when they appear first in the main text. It is the case (at least) for "ZFC" and "H_s".

3) The definition of critical current given in Methods ("Data analysis") is not clear/well defined. The authors wrote:

"Data Analysis. In Fig. 3a, where we present the asymmetry factor ΔI_c , I_c is defined as the lowest current in which we measure an onset of voltage and is obtained by setting a threshold just above the noise level for the derivative of the curves presented in Fig. 2a-b"

First of all, the chosen thresholds must be clearly reported (in Methods), and compared explicitly to the noise level (a graph would be helpful, at least in the Supp Info). It is not clear to me why the threshold applies to the *derivative* and not to the V values themselves.

4) Possibly related to the previous point: the IVs in Fig 2 are "S-shaped", i.e., there is a finite slope (resistance) already at very small bias, an indication that nonlinear dissipation from vortex motion (creep) is already at work. What distinguishes the low-bias regime from that after the discontinuity in the slope (the one the authors seem to take as critical current)?

5) The small black arrows in Fig 2a and 2b should indicate, if I understood it correctly, that the IVs are swept always from zero bias to finite (positive or negative) bias, to avoid heating effects producing hysteresis. Is it so? If yes, I would make the caption of this Figure more explicit about this detail.

6) The authors write:

"If the vortices enter with more ease from one edge compared to the other, it implies that the critical current in the positive direction should be different from that in the negative direction ($|I+c| \neq |I-c|$)."

What about the barrier to *leave* the sample? What about pinning? I might agree with the statement of the authors (i.e., that the barrier to enter the sample is the most relevant quantity), but it needs to be compared to the barrier to leave the sample and to the pinning strength. Otherwise, it might seem that a vortex that enters will always, with certainty, travel across and leave the sample (generating dissipation), irrespective of pinning and the other edge barrier.

7) Line 220: "For H_y, 3 states..."

The authors probably mean something like (I use latex notation here): "For $\vec{H} \parallel \hat{y}$ ".

8) Line 262: "... a gauge term that imposes... ": in the following integral $j_s(y)$ is a scalar but in Eq(1), it is a vector. It must be made explicit which component is meant.

9) Line 272: The parameter a (and the entire Eq 2) is taken from Zeldov et al., Ref[29]. I think it is important to state explicitly what the physical meaning of a is, i.e. the central field-free region. It would be useful to show this in the graphs on the right side of Fig.4.

10) Very important: the auxiliary lines (orange) in Fig 4a are not discussed in the text. This must be done. I would improve the connection between text and figure, especially the panel a. As it is now, it is not very clear.

-> In Fig4b and 4c I would indicate the value of the parameter I_t/I_c (which also determines "a") used.

Reviewer #1 (Remarks to the Author):

The idea of direct imaging a superconducting vortex diode effect is very interesting. However, the present results are not convincing. This is because there are no images showing clear vortices and/or vortex flow channels. In contrast, unambiguous vortices and vortex flow channels were demonstrated in Ref. 30 which was published by one of the same corresponding authors and using the same imaging technique. I guess it could be difficult to directly visualize the vortex flow channels on the present sample with a magnetic layer on top of a superconducting film. The EuS could be a magnetic shielding layer that would prevent the accessing of the magnetic signals from vortices. I suggest the authors to repeat the experiments on a sample in which the magnetic EuS layer is under the superconducting film. In this case, the vortices and the diode-like vortex flow could be directly visualized.

We thank the reviewer for noting that the work is “very interesting”, and for raising helpful suggestions to improve the manuscript.

Regarding the reviewer's suggestion of reversing the order of the layers, there are fundamental reasons why this would be impractical:

1. The superconducting Nb layer is only 8-nm-thick – hence reversing the order of Nb and EuS would expose the superconducting Nb to air which would oxidize the film. We could in principle cap the Nb with, for example, a layer of Au or Pt but this would dramatically change the superconducting properties through the proximity effect (suppressing T_c).

2. To grow a homogeneous 8-nm-thick layer of Nb requires growth on an atomically flat substrate to minimize roughness, particularly at the Nb/EuS interface. Growing Nb on EuS would have increased the roughness of Nb to several nanometers over a lateral length of only several micrometers, adding an uncontrolled effect to the electrical properties of the Nb.

In addition, we are not aware of a mechanism by which a highly resistive ferromagnet like EuS could screen a time-dependent magnetic field. In conducting metals, it is known that screening currents are present in the case of a time-dependent magnetic field (Lenz's law). DC screening (diamagnetism) or AC screening are excluded for the case of a ferromagnetic insulator, or greatly reduced for poorly conducting ferromagnets.

The referee rightfully points out that our images exhibit different features from the ones shown in Ref. 30 (now Ref. 33). We think this is due to two reasons:

1. In Ref. 33, a DC current was applied, whereas in the present work, we apply an AC current. We had to apply an AC current and measure the oscillating component of the magnetic field $B_z^{ac}(x, y)$ to filter out the DC magnetic component generated at the EuS edges. We emphasize that the static stray field caused by the EuS can be up to two orders of magnitude larger than the time-dependent component of the field. In Ref. 33 we did not have any magnetic layer, therefore, DC measurements were possible. In the AC measurements, the flux flow is reinitialized at each cycle, and we average over many cycles for each image pixel. Given the vortex routes might be different in every initialization, this causes smearing of the signal. Thus, individual channels are harder to resolve.

2. The Pearl length in both experiments is also radically different. In Ref. 33 the sample is a Pb film with thickness $d = 75$ nm and $\lambda = 96$ nm [Embon *et al.* Sci. Rep. 5, 7598 2015]. Those parameters imply that the magnetic size of the vortex, given by the Pearl length, is $\Lambda = \frac{\lambda^2}{d} = 122$ nm $\sim \lambda$. In contrast, in our current work, $\lambda \sim 230$ nm [Gubin *et al.* 72, 064503 (2005)] and the thickness of the Nb $d = 8$ nm resulting in $\Lambda = 6$ μ m. Thus, the Pearl length in the current work is more than an order of magnitude larger than in Ref. 30 (now Ref. 33). Having a larger vortex implies that the same magnetic flux is spread over a larger area, resulting in weaker fields that are harder to detect. This further smears the vortex signal, hampering the observation of single vortex channels.

Following the reviewer's comment, we clarify the above on page 3, lines 127-137, and reproduced below:

“Given that we average over an ac signal with a time scale many orders of magnitude larger than those involved in the vortex dynamics, our images effectively portray the density of vortex paths across the device. For that reason, the signal along the channels becomes undetectable in the inner part of the sample. We emphasize that we are observing vortices in a thin film superconductor with an effective penetration depth $\Lambda \sim 6\mu\text{m}$. This length scale defines the effective size of the vortex. Having a larger vortex implies that the same magnetic flux is spread out over a larger area, resulting in weaker fields that are harder to detect. This further smears the vortex signal, hampering the observation of single vortex channels.”

However, despite the differences between the images of this work and the ones in Ref. 30 (now Ref. 33), there are important similarities that indicate that vortices are involved in our work. First, the magnetic field is larger at the entry point. Second, these features appear simultaneously with the voltage onset, indicating the involvement of flux flow.

Reviewer #2 (Remarks to the Author):

This is a very thorough and impressive experimental work showing how a ferromagnetic layer modifies the transport properties of a superconductor through its influence on vortices (fluxes). The results are of broad interest given the current trends in superconducting electronics for energy efficient and quantum information technologies. One aspect, I believe, is not described clearly in the paper: what is the behaviour of the magnetic layer without the superconductor? Which images were taken above T_c (if any)? If there are no such data, can the authors describe the expected behavior of the FM layer only?

We thank the reviewer for their positive comment.

Regarding measurements above T_c , the referee raises exciting points. Unfortunately, it is currently impossible for us to perform SOT microscopy much above 4.2 K. The reason is that we are limited by the T_c of the Pb, which is used in the fabrication of the SOT. The T_c of the SOT is lower than the bulk Pb (7.2 K), which is not much larger than the T_c of our sample, 5.6 K. Since the sensitivity of the SOT is largely reduced close to T_c , we cannot perform measurements above 5.6 K. We note that it is unlikely that the presence of the thin Nb film (8 nm) will influence the magnetic behavior of the EuS. The evidence resides in the global $M(H)$ and $M(T)$ measurements. We measured two $M(H)$ loops, one above T_c and one below (see below, panel a). Besides a small difference in saturation magnetization due to the change in temperature, no significant features differentiate the two states. We note that the EuS has a Curie temperature of about $T_{Curie} = 20$ K, as demonstrated by the $M(T)$ data acquired by applying a field of 500 G (see below panel b). No magnetic transition is found at T_c . Following the reviewer's comment, we added the figure shown below to the supplementary information, together with the following remark in the main text (page 2, lines 76 – 81):

“We assume, there is no effect on the magnetic properties of EuS due to the coupling to superconducting Nb. We confirm this assumption by performing global magnetization measurements on an unpatterned EuS/Nb device with identical thicknesses and show that there is a negligible difference across the SC transition temperature (supplementary fig. 1)”

Reviewer #3 (Remarks to the Author):

The present manuscript describes combined scanning SQUID and transport experiments in superconductor/ferromagnet bilayers. The goal is to elucidate the mechanism leading to the emergence of non-reciprocal supercurrent (superconducting diode effect) in ordinary superconductors when placed on ferromagnets.

On the one hand, this mechanism is here demonstrated to be pretty conventional, as anticipated by other works (Ref.[17], but also by [D. Suri et al., Applied Physics Letters 121, 102601 (2022).]). In addition, the experimental technique is not totally new. Still, the application of this imaging technique to this particular problem helps to clarify directly and beyond any doubt the source of non-reciprocal supercurrent in S/F bilayers. Owing to the great interest in this topic by the scientific community, I find the manuscript potentially suitable for publication in Nature Communications, provided that the questions below are addressed.

We thank the reviewer for their constructive comments, and for acknowledging the potential relevance of our work.

1) The authors write in the introduction: "This type of symmetry breaking has been observed in S/F bilayers as a non-reciprocal critical current [15–19] and..."

The authors here are talking about a system with Rashba SOC with additional exchange interaction (see beginning of the paragraph). The references 18-19 appear therefore misplaced in this context, since there is no SF bilayer there. If, instead, the goal of the authors is simply to refer to the recent observations of spin-perturbation induced SC diode effect (which I would find appropriate for this paper), then 18-19 are fine but then I would, for consistency, also add two relevant works that appeared before Ref 18, namely:

-C. Baumgartner, et al., Nature Nano. 17, 39 (2022).

-H. Wu et al., Nature, 604, 653 (2022).

Also, given the topic treated in this manuscript, I would recommend to cite the aforementioned reference [D. Suri et al., Applied Physics Letters 121, 102601 (2022).]

We thank the reviewer for suggesting these relevant references. Following the reviewer's comment, the suggested references have been added, and the mentioned references (22-24) have been moved to a more appropriate location in the text.

2) Sometimes symbols and acronyms are defined first in Methods and not when they appear first in the main text. It is the case (at least) for "ZFC" and "H_s".

We have carefully checked the manuscript for undefined variables and acronyms, and have ensured these are all now defined.

3) The definition of critical current given in Methods ("Data analysis") is not clear/well defined. The authors wrote:

"Data Analysis. In Fig. 3a, where we present the asymmetry factor ΔI_c , I_c is defined as the lowest current in which we measure an onset of voltage and is obtained by setting a threshold just above the noise level for the derivative of the curves presented in Fig. 2a-b"

First of all, the chosen thresholds must be clearly reported (in Methods), and compared explicitly to the noise level (a graph would be helpful, at least in the Supp Info). It is not clear to me why the threshold applies to the *derivative* and not to the V values themselves.

We thank the reviewer for pointing out this omission. We first want to emphasize that the non-reciprocal behavior of $I(V)$ is clearly visible in the raw data shown in Fig. 2a. In Fig. 3a, we solely want to follow the general behavior as a function of the applied field and compare it with the $M(H)$ curve. Although the exact values presented in Fig. 3a are criterion-dependent, our message is the same - that the non-reciprocity follows the magnetization i.e., $\Delta I_c > 0$ for $M > 0$, $\Delta I_c < 0$ for $M < 0$ and $\Delta I_c \sim 0$ for $M \sim 0$. This is reinforced by our $B_z^{ac}(x, y)$ and $B_z^{dc}(x, y)$ images. Nevertheless, we completely agree with the reviewer that the explanation of how the critical current is determined was missing in the previous version of our manuscript. For the analysis in Fig. 3a, we used the derivative criterion because it gave lower noise in I_c . The derivative criterion also has the advantage that it is insensitive to the DC offset. We set our threshold to 20 m Ω . This is justified by looking at the deviation at low current (see supplementary Fig. 2). Determining I_c by setting it above the noise level around 2 μ V gives similar results to Fig. 3a. Following the reviewer's comments, we clarify our method of determining I_c in the method section (see below), and reproduced below:

"To determine this point, we take the derivative of $V(I)$ and impose a threshold above the noise level determined to 20 m Ω . Supplementary figure 2 shows the derivative of a typical $V(I)$ curve with the corresponding threshold of 20 m Ω that was used to obtain the data presented in Fig. 3a. It is also possible to perform this analysis by assuming a threshold on the voltage value of the curves. For this method we impose a threshold of 2 μ V. We obtain similar results using this criterion as shown in supplementary figure 3."

Furthermore, we have added in the supplementary information a figure to justify our method. Finally, we also added to the supporting information a plot of $\Delta I_c(H)$ based on the voltage criterion suggested by the reviewer.

4) Possibly related to the previous point: the IVs in Fig 2 are "S-shaped", i.e., there is a finite slope (resistance) already at very small bias, an indication that nonlinear dissipation from vortex motion (creep) is already at work. What distinguishes the low-bias regime from that after the discontinuity in the slope (the one the authors seem to take as critical current)?

We thank the reviewer for pointing out this feature. As we discussed in our reply to the previous point, for the purpose of the analysis in Fig. 3a we do not make any distinction between the two regimes, we simply use the criterion $\frac{dV}{dI} > 20$ m Ω as a means to quantify the asymmetry. We consider that the clarifications made in the previous point also address this point.

5) The small black arrows in Fig 2a and 2b should indicate, if I understood it correctly, that the IVs are swept always from zero bias to finite (positive or negative) bias, to avoid heating effects producing hysteresis. Is it so? If yes, I would make the caption of this Figure more explicit about this detail.

We thank the reviewer for pointing out this missing detail. The reviewer is correct. The critical current can be different when crossing from the superconducting state (switching current in the context of Josephson junctions) or from the voltage state (retrapping current). As the reviewer pointed out, this could be caused by heating. Following the reviewer's recommendation, we added this detail to the caption and to the main text.

6) The authors write:

"If the vortices enter with more ease from one edge compared to the other, it implies that the critical current in the positive direction should be different from that in the negative direction ($|I_{+c}| \neq |I_{-c}|$)." What about the barrier to *leave* the sample? What about pinning? I might agree with the statement of the authors (i.e., that the barrier to enter the sample is the most relevant quantity), but it needs to be compared to the barrier to leave the sample and to the pinning strength. Otherwise, it might seem that a vortex that enters will always, with certainty, travel across and leave the sample (generating dissipation), irrespective of pinning and the other edge barrier.

We thank the reviewer for highlighting this point. Dominant pinning would oppose the diode effect and thus prevent us from seeing its signature in electrical transport and in SOT microscopy. Our data suggest that once the barriers to enter and to leave are overcome, there is flux flow. Regarding the barrier to leave the sample, the referee is right; both barriers need to be overcome to reach flux flow conditions. We note that the Meissner screening currents caused by the EuS has the effect of lowering both barriers. For a given saturated magnetization, vortices are repelled from the entry point and attracted by the exit point. That is what favors one flux flow direction. Following the reviewer's comment we clarify these two points in page 8, lines 342-352:

“Moreover, apart from the claim that the entry barrier of the sample is overcome, another condition that needs to be fulfilled is that the barrier to leave the sample must be overcome as well. Our data suggests that the Meissner screening currents caused by the EuS has the effect of lowering both barriers. For a given saturated magnetization, vortices are repelled from the entry point and attracted by the exit point. That is what favors one direction of vortex flow. Furthermore, the existence of the diode effect, as observed both in transport and in SOT microscopy, contradicts the dominance of pinning in the bulk of the sample, as pinning should suppress the possibility of flux-flow, regardless of the entry and exit barriers.

7) Line 220: "For H_y , 3 states..."

The authors probably mean something like (I use latex notation here): "For $\vec{H} \parallel \hat{y}$ ".

We thank the reviewer for noticing this issue. We have added a clarification of the notation used in the main text on page 3, line 127.

8) Line 262: "... a gauge term that imposes...": in the following integral $j_s(y)$ is a scalar but in Eq(1), it is a vector. It must be made explicit which component is meant.

We thank the referee for highlighting this critical point. We have modified the integral in eq. 1 so that it is now clear that the intention is to integrate the x component of the vector $j_s(y)$.

9) Line 272: The parameter a (and the entire Eq 2) is taken from Zeldov et al., Ref[29]. I think it is important to state explicitly what the physical meaning of a is, i.e. the central field-free region. It would be useful to show this in the graphs on the right side of Fig.4.

Following the reviewer's comment, we have clarified the meaning of this parameter. Furthermore, we have added a visual marking of the field-free region in Fig. 4b.

10) Very important: the auxiliary lines (orange) in Fig 4a are not discussed in the text. This must be done. I would improve the connection between text and Figure, especially the panel a. As it is now, it is not very clear.

-> In Fig4b and 4c I would indicate the value of the parameter I_t/I_c (which also determines "a") used.

We thank the reviewer for pointing out this omission. Due to this important remark, we have added an explanation of the magnetic sources used to derive A_M (orange auxiliary lines) in the text:

“The corresponding vector potential \vec{A}_M can be derived from the magnetostatic equivalent of the Poisson equation, assuming the magnetic sources are 2 infinite auxiliary wires located on the edges of the F layer”

Furthermore, we have stated the value of $\frac{I_t}{I_c} = 0.8$ in the figure caption.

REVIEWERS' COMMENTS

Reviewer #1 (Remarks to the Author):

First, the vortex diode effect and the superconducting diode effect are two different phenomena. The first emphasizes the ratchet motion of vortices which is an effect from superfluid of Cooper pairs (although the motion of vortices dissipate energy leading to small resistance). The nonreciprocal critical currents of vortex diode effects are determined by vortex 'depinning' currents. While the superconducting diode effect [first unambiguously demonstrated in Nature 584, 373 (2020)] emphasizes on nonreciprocal superconducting-to-normal transition. The nonreciprocal critical currents are defined by 'depairing' currents. In some cases, a very strong vortex ratchet effect can induce a superconducting diode effect [e.g. Nature Communications 12, 2703 (2021)].

Although the vortex diode effect (or ratchet effect) has been widely investigated more than two decades ago, a direct imaging of the unidirectional vortex motion has never been demonstrated. Thus, a direct visualization of such effect is important to both fundamental and applied researches of superconductivity. Therefore, this work certainly deserves to be published in a high impact journal.

What is not perfect is that vortices and/or vortex flow channels were not directly visible (mostly due to the order of sample layers). This makes the imaging of vortex diode effect in an indirect way. Unfortunately, the superconducting and magnetic layers seem difficult to be reversed (The authors listed several possible technical challenges for reversing the order of the layers in their response to my last comments), which prevents directly observing vortices, but even so I found the imaging, the transport data as well as the theoretical analysis were flawless. Below I list several minor points for the authors to consider.

1. To extract the signal of vortex flow over the very large magnetic signal of the top EuS layer, the authors applied an ac imaging technique. From my understanding (please correct me if I am wrong) the successful application of the ac imaging is based on a critical effect 'bifurcation of vortex flow channels'. This leads to the vortex density of vortex-flow channels near the edge is higher than those at the center. In contrast the vortex density would be nearly identical from edge to center when vortices not flowing. This leads to the vortex density near the edge when vortices flowing is more inhomogeneous than that when vortices not flowing. The ac imaging captures the change of vortex density near the edge under an ac current driven. Therefore, the channel bifurcation is crucial to the ac imaging. The bifurcation of vortex flow channels was clearly demonstrated previously by some of the same authors in Ref. 33. In Ref. 33, the sample is a superconducting stripe with a constriction, which results in nonuniform current density along the stripe. The vortex penetration and the bifurcation of vortex-flow channels are induced by the stripe constriction. In this work, the sample stripe is uniform with parallel edges with uniform current density along the stripe. One would expect there was no bifurcation of vortex channels in an ideal stripe sample. However, in a practical sample the bifurcation could exist (e.g., due to existing of defects or sample nonuniform). Would it be possible to demonstrate the bifurcation in a uniform stripe (e.g. using TDGL simulation)? Or could the authors provide some discussions of sample geometry and/or any other factors on the bifurcation effect?

2. The abbreviation 'OOP' in page 2 is not defined when first using.

3. What's the value of the ac current used for SOT imaging? Is there an ac current dependence of the SOT images at $I_{ac} > I_c$?

4. Since the ac imaging technique is used here, would it be better to show the rectification dc voltage signal (V_{dc} versus I_{ac}) and compare it to the SOT images?

5. In the Discussion paragraph, the authors state 'the predicted non-reciprocal critical current that triggers GHz vortex flow is observed in both local imaging techniques and global transport measurements.' Could the authors provide the data/analysis or cite any corresponding references to this statement?

Reviewer #3 (Remarks to the Author):

The authors satisfactorily and thoroughly answered the Referees' questions. I therefore recommend proceeding with publication.

Reply to Reviewer

Reviewer #1 (Remarks to the Author):

First, the vortex diode effect and the superconducting diode effect are two different phenomena. The first emphasizes the ratchet motion of vortices which is an effect from superfluid of Cooper pairs (although the motion of vortices dissipate energy leading to small resistance). The nonreciprocal critical currents of vortex diode effects are determined by vortex 'depinning' currents. While the superconducting diode effect [first unambiguously demonstrated in Nature 584, 373 (2020)] emphasizes on nonreciprocal superconducting-to-normal transition. The nonreciprocal critical currents are defined by 'depairing' currents. In some cases, a very strong vortex ratchet effect can induce a superconducting diode effect [e.g. Nature Communications 12, 2703 (2021)].

Although the vortex diode effect (or ratchet effect) has been widely investigated more than two decades ago, a direct imaging of the unidirectional vortex motion has never been demonstrated. Thus, a direct visualization of such effect is important to both fundamental and applied researches of superconductivity. Therefore, this work certainly deserves to be published in a high impact journal.

What is not perfect is that vortices and/or vortex flow channels were not directly visible (mostly due to the order of sample layers). This makes the imaging of vortex diode effect in an indirect way. Unfortunately, the superconducting and magnetic layers seem difficult to be reversed (The authors listed several possible technical challenges for reversing the order of the layers in their response to my last comments), which prevents directly observing vortices, but even so I found the imaging, the transport data as well as the theoretical analysis were flawless. Below I list several minor points for the authors to consider.

We thank the reviewer for the positive comments acknowledging the relevance of our work.

1. To extract the signal of vortex flow over the very large magnetic signal of the top EuS layer, the authors applied an ac imaging technique. From my understanding (please correct me if I am wrong) the successful application of the ac imaging is based on a critical effect 'bifurcation of vortex flow channels'. This leads to the vortex density of vortex-flow channels near the edge is higher than those at the center. In contrast the vortex density would be nearly identical from edge to center when vortices not flowing. This leads to the vortex density near the edge when vortices flowing is more inhomogeneous than that when vortices not flowing. The ac imaging captures the change of vortex density near the edge under an ac current driven. Therefore, the channel bifurcation is crucial to the ac imaging. The bifurcation of vortex flow channels was clearly demonstrated previously by some of the same authors in Ref. 33. In Ref. 33, the sample is a superconducting stripe with a constriction, which results in nonuniform current density along the stripe. The vortex penetration and the bifurcation of vortex-flow channels are induced by the stripe constriction. In this work, the sample stripe is uniform with parallel edges with uniform current density along the stripe. One would expect there was no bifurcation of vortex channels in an ideal stripe sample. However, in a practical sample the bifurcation could exist (e.g., due to existing of defects or sample nonuniform). Would it be possible to demonstrate the bifurcation in a uniform

stripe (e.g. using TDGL simulation)? Or could the authors provide some discussions of sample geometry and/or any other factors on the bifurcation effect?

We thank the reviewer for highlighting this important point. We agree with the reviewer that bifurcation plays an important role with the signal that we see. Bifurcations randomize the vortex trajectory, which in turns lowers the contrast seen with the SOT. While the entrance point is usually determined by defects at the edge that are fixed in space, the bifurcation depends on fixed defects but also on the position of other vortices at a given time. The geometrical constriction in the sample of Ref. 33 (Ref. 39 in the revised version) allowed us to limit the number of entry points to just a few. In other words, the constriction allows us to modulate the current density along the direction of the current (x axis in current article and y axis in Ref. 33). The bifurcation has another physical origin that does not require the geometrical constriction. The bifurcation originates from the non-uniform current distribution along the direction *perpendicular* to the current (y axis in current article and x axis in Ref. 33). That was theoretically derived by Zeldov *et al.* in Ref. 32 of the present article. The non-uniform current distribution originates from the Meissner screening currents that exist in any superconducting thin film. The current distribution for our geometry is plotted in figure 4b,c. We note that the current density is larger at the edges. This implies that a vortex enters the sample with a higher velocity and slows down as it penetrates into the sample where the current density is lower. As a result, as the vortex goes towards the center, the vortex-vortex distance decreases, as shown in Fig. 4a of Ref. 33. At some point, the vortex-vortex distance becomes too small and the mutual repulsive force causes the vortices to bifurcate. Moreover, defects in the sample can also cause bifurcation as the reviewer rightfully pointed out. Once the vortex passes the center of the sample, it accelerates towards the other edge of the sample. The exit point is randomized by the bifurcation that previously occurred. As the reviewer suggested, this effect can indeed be visualized in a movie resulting from TDGL simulations published in Ref. 33 (see movies 5 and 6).

To summarize the above paragraph, the geometrical constriction is not necessary to have bifurcation. Moreover, in Ref. 33 we argue that the geometrical constriction gives rise to a temperature profile that tends to *prevent bifurcation*. This idea is captured in Fig. 4d of Ref. 33 where we show that the critical vortex-vortex distance at which the bifurcation occurs goes down with increasing dissipation. Intuitively, having a geometrical constriction gives a hot spot centered in the center of the constriction. Vortices are confined by the hot spot because it cost less energy to locate a vortex in an area closer to T_c . By contrast, having a straight Hall bar yields a temperature profile that is rather uniform along the Hall bar and the thermal confinement is expected to vanish.

Finally, we wish to clarify that the non-uniform current distribution $J_x(y)$ causes the vortex density to be larger around the center of the sample. The larger vortex density also reduces the signal since the vortices start to overlap and the field modulation decreases.

Following the reviewer's comment, we further clarify that point in the article on page 4. We added the following text:

“Vortex channels are observed only along the sample edges due to bifurcations that occur along their path, which randomize their trajectories. The bifurcation originates from the non-uniform current distribution that modulates along the y axis [38]. The vortex enters the sample with a higher velocity due to the larger current density along the edge and slows down as it penetrates into the sample where the current density is lower. Finally, as the vortex flows towards the center,

the vortex-vortex distance decreases and the mutual repulsive force causes the vortices to bifurcate[39].”

2. The abbreviation ‘OOP’ in page 2 is not defined when first using.

We thank the reviewer for pointing that omission on our side. The definition was added on page 2.

3. What’s the value of the ac current used for SOT imaging? Is there an ac current dependence of the SOT images at $I_{ac} > I_c$?

We thank the reviewer for highlighting this point. The images shown in figure 2 and 3 were acquired with $I_{ac} = 0.42$ mA (RMS). Above I_c , we see that the signal of the image grows linearly until we reach a value where the dissipation is too high and the sample becomes entirely normal. Following the reviewer’s comment, we discuss this on page 4. We also specify the current values in the figure captions.

4. Since the ac imaging technique is used here, would it be better to show the rectification dc voltage signal (V_{dc} versus I_{ac}) and compare it to the SOT images?

We thank the reviewer for the suggestion. Indeed, from an engineering point of view it would make more sense to show V_{dc} vs. I_{ac} . We believe that the $I - V$ characteristics in dc gives equivalent information showing clearly the diode effect. Measuring V_{dc} vs. I_{ac} at this point would require a lot of work and would not add new fundamental information to the article. We note that from the engineering viewpoint, a lot of device optimization is required. One could then think about showing the rectification voltage as a function of the different device geometry, but we think that this is beyond the scope of our present article. For that reason, we respectfully ask the reviewer to agree to publish with the current data set.

5. In the Discussion paragraph, the authors state ‘the predicted non-reciprocal critical current that triggers GHz vortex flow is observed in both local imaging techniques and global transport measurements.’ Could the authors provide the data/analysis or cite any corresponding references to this statement?

We thank the reviewer for pointing out this missing analysis. The voltage generated per vortex channel is $V = \Phi_0 f$, where f is the vortex entrance frequency and Φ_0 is the flux quantum, which can be written as $\frac{1}{4.8} \times 10^{-14}$ V/Hz. From our images, we estimate that there are roughly 0.3 vortex channels per micron, giving a total of 20 channels between the voltage electrodes. Our transport measurements output a voltage on the order of 40 μ V which translates into a frequency of 9.6×10^8 Hz. Following the reviewer’s comment, we clarify this point in our discussion of the revised manuscript.